# Assessment of the Chad guinea worm surveillance information system: A pivotal foundation for eradication

Saugat Karki[1]*, Adam Weiss[2], Jina Dcruz[3], Dorothy Hunt[2], Brandon Haigood[2], Philip Tchindebet Ouakou[4], Elisabeth Chop[2], Hubert Zirimwabagabo[2], Beth L. Rubenstein[5,6], Sarah Yerian[2], Sharon L. Roy[7], Mary L. Kamb[7], Sarah Anne J. Guagliardo[2,7]

1 Surveillance and Data Management Branch, Division of STD Prevention, Centers for Disease Control and Prevention, Atlanta, Georgia, United States of America, 2 Guinea Worm Eradication Program, The Carter Center, Atlanta, Georgia, United States of America, 3 Population Health Workforce Branch, Division of Scientific Education and Professional Development, Centers for Disease Control and Prevention, Atlanta, Georgia, United States of America, 4 Guinea Worm Eradication Program, Ministry of Public Health and National Solidarity, N'Djamena, Chad, 5 Epidemic Intelligence Service, Centers for Disease Control and Prevention, Atlanta, Georgia, United States of America, 6 Malaria Branch, Division of Parasitic Diseases and Malaria, Centers for Disease Control and Prevention, Atlanta, Georgia, United States of America, 7 Parasitic Diseases Branch, Division of Parasitic Diseases and Malaria, Centers for Disease Control and Prevention, Atlanta, Georgia, United States of America

* SKarki@cdc.gov

## Abstract

### Background

In the absence of a vaccine or pharmacological treatment, prevention and control of Guinea worm disease is dependent on timely identification and containment of cases to interrupt transmission. The Chad Guinea Worm Eradication Program (CGWEP) surveillance system detects and monitors Guinea worm disease in both humans and animals. Although Guinea worm cases in humans has declined, the discovery of canine infections in dogs in Chad has posed a significant challenge to eradication efforts. A foundational information system that supports the surveillance activities with modern data management practices is needed to support continued program efficacy.

### Methods

We sought to assess the current CGWEP surveillance and information system to identify gaps and redundancies and propose system improvements. We reviewed documentation, consulted with subject matter experts and stakeholders, inventoried datasets to map data elements and information flow, and mapped data management processes. We used the Information Value Cycle (IVC) and Data-Information System-Context (DISC) frameworks to help understand the information generated and identify gaps.

### Results

Findings from this study identified areas for improvement, including the need for consolidation of forms that capture the same demographic variables, which could be accomplished

**Data Availability Statement:** No quantitative data was used in the manuscript, therefore we have no datasets. Some institutional information used in the evaluation regarding The Carter Center, or Chad

Guinea Worm Eradication Program can be obtained by contacting info@cartercenter.org.

**Funding:** This work was supported by The Carter Center, whose work to eradicate Guinea worm disease has been made possible by financial and in-kind contributions from many donors. A full listing of supporters can be found at The Carter Center website (http://www.cartercenter.org/ donate/corporate-government-foundation-partners/index.html). The funders had no role in study design, data collection and analysis, decision to publish, or preparation of the manuscript.

**Competing interests:** The authors have declared that no competing interests exist.

with an electronic data capture system. Further, the mental models (conceptual frameworks) IVC and DISC highlighted the need for more detailed, standardized workflows specifically related to information management.

## Conclusions

Based on these findings, we proposed a four-phased roadmap for centralizing data systems and transitioning to an electronic data capture system. These included: development of a data governance plan, transition to electronic data entry and centralized data storage, transition to a relational database, and cloud-based integration. The method and outcome of this assessment could be used by other neglected tropical disease programs looking to transition to modern electronic data capture systems.

## Author summary

Guinea worm disease has no pharmacological treatment or vaccines, and therefore existing prevention and control strategies (e.g., case containment, health education, chemical treatment of water bodies) are critically dependent on timely, accurate, and actionable data. We conducted informant interviews, used conceptual frameworks, and mapped data flow to evaluate the Chad Guinea Worm Eradication Program's current information system. We identified areas for improvement including the need to consolidate variables across data collection forms and the need to develop streamlined workflows. We proposed a four-phased roadmap for transitioning to an electronic data capture system and centralizing data storage. Our approach and proposed roadmap could be adopted by other neglected tropical disease control programs looking to modernize data collection and storage procedures.

## Introduction

### Public health context

Guinea worm disease is targeted for global eradication [1] and is caused by infection with the parasitic nematode *Dracunculus medinensis*. Guinea worm emergence from the body is painful. This pain, along with secondary infections of the wound, can cause disability, preventing afflicted individuals from doing daily work. This disability often lasts for about two months but can be permanent [2,3]. The discovery of canine infections in dogs in Chad has posed a significant challenge to eradication efforts [4,5].

The Republic of Chad reports the greatest number of Guinea worm cases worldwide: in 2020, there were 12 human cases, 1,507 canine cases, and 61 feline cases reported from Chad [6]. There is no vaccine or pharmacological treatment for Guinea worm disease, and accordingly, eradication efforts rely on containment (i.e., preventing water contamination), treatment of appropriate water resources suspected to be contaminated with Abate (temephos), and health education as important tools to prevent transmission. The Chad Guinea Worm Eradication Program (CGWEP) uses data to inform these prevention and control strategies, and to create periodic and ad hoc reports that are disseminated to partners and stakeholders. Accurate, timely, high quality, actionable data are therefore the foundation for effectively informing these strategies.

## Public health informatics context

Technological advances, including data management tools (e.g., cloud-based systems, relational databases), software programs (e.g., Python, R, Microsoft Access), information systems (e.g., surveillance systems, data registries), and better data management and analytics strategies (e.g., deduplication, data visualization) have presented us with the opportunity to link important data at an unprecedented pace and scale. Technology to support scalable public health surveillance systems and processes to effectively manage and use information collected have become financially viable and relatively easier to use [7]. Modernizing information systems in the neglected tropical diseases (NTDs) realm has great potential to inform disease control programs [8]. Furthermore, 'Public Health 3.0', a call to action for modern public health, recognized timely, reliable, granular-level, and actionable data as a foundational recommendation to meet the challenges of the 21st century [9].

Previous works comparing electronic data collection to paper-based data collection for NTDs have shown that data collected through electronic sources were less expensive, more efficient, and produced higher quality data with fewer errors [10,11]. However, there are few studies describing digital health technologies in surveillance, management, and treatment of NTDSs [12]. Although some studies reported on enhancements of the surveillance and management of NTDs, they were limited to the description of specific tools used for those processes [12–14].

## Objective

At the time of this investigation, Guinea worm surveillance data were collected in a siloed manner with inefficiencies in data management processes. For example, information collected regarding the same Guinea worm case was not linked because data were collected in isolation via paper-based surveys. Although the existing information system worked well in the past, as evidenced by the 99.9% reduction in human Guinea worm cases since the mid-1980s [1,15,16], in the context of eradication, identifying and containing every single case becomes increasingly difficult as cases decline [17,18]. At the same time, the number of Guinea worm "rumors" (unverified reports of possible cases) in Chad has increased substantially. For example, Guagliardo et al (2020) reported that there were 504 dog rumors in 2015 compared to 15,511 rumors in 2018 [4]. The enormous increase in rumors over time has underscored the need for improved information systems to facilitate detection, tracking, and containment.

To address the data and information system-related challenges, the Chad Ministry of Public Health and National Solidarity (MoPH) and The Carter Center requested informatics assistance from the Centers for Disease Control and Prevention (CDC) in the form of an "Info-Aid," a mechanism that allows Public Health Informatics Fellows (CDC's "data detectives") to provide short-term technical assistance in the event of an urgent public health need for applied information science and technology [19]. Although a previous, rigorous surveillance evaluation had been conducted [20], it was determined that an additional assessment would be needed specifically to address informatics needs.

The purpose of this report is to describe the approach and methods used to in the Info-Aid, and to outline a roadmap to improve CGWEP information systems' data flow, upstream and downstream data management practices, and system inefficiencies. Previously published studies revealed a gap in a systems-level assessment of information systems unique to NTDs [10], and therefore, this approach could also be adopted by other NTD programs [21].

## Methods

### Ethics statement

All respondents were asked for verbal consent before proceeding with the interviews about data quality. This project was given a non-research determination by the delegated authority at the Center for Global Health, Centers for Disease Control and Prevention (protocol #: 0900f3eb819fb155).

### Surveillance system

CGWEP is run by the Chad MoPH with support from The Carter Center and operates in approximately 2,300 villages throughout Southern Chad. Surveillance is simultaneously conducted in both humans and domestic animals (primarily dogs) [22], and involves both active and passive surveillance, described in detail elsewhere [4]. The data collected by the surveillance system are owned by the government of Chad and have been used previously to discern epidemiologic trends in humans and dogs alike [4,23].

### Information gathering and data quality interviews

We gathered relevant literature and documentation regarding the informatics landscape and CGWEP operations. Although there were data dictionaries for some surveys, no program-level metadata and data dictionary existed.

To document the public health context, existing information systems (people, process, and technology), and informatics need and gaps, we consulted with staff from CGWEP and The Carter Center in N'Djamena and Atlanta during August and September of 2019 (Table 1).

**Table 1. Description of roles of staff consulted and contributions to Info-Aid outcomes.**

| Role within organization | Number of staff consulted | Contributions to Info-Aid Outcomes |
| --- | --- | --- |
| Epidemiologist (The Carter Center Headquarters) | 1 | • Organizational structure<br>• How data is used<br>• Mapping of survey forms and data elements<br>• Data dictionaries, context for use of data |
| Associate Directors (The Carter Center Headquarters) | 3 | • Organizational structure<br>• Context for use of data<br>• Use and intention of survey forms<br>• Programmatic needs<br>• Future direction |
| Information Technology Staff (The Carter Center Headquarters) | 3 | • Introduction to technology used (ELMO-NEMO)<br>• Realistic technological capabilities, potential, and limitations<br>• Development of survey instruments in ELMO-NEMO |
| Data Manager (CGWEP[a]) | 1 | • Data and information flow mapping<br>• Quantification of burden for data entry and compilation<br>• Documentation of data storage and management practice |
| Data Entry Staff (CGWEP[a]) | 5 | • Time taken for data entry and data collection<br>• Identification of process bottlenecks<br>• Assessment of the need for institutionalizing processes<br>• Assessment of the need to consolidate processes and centralize data storage |
| Technical Advisors (CGWEP[a]) | 3 | • Time taken for data entry and collection<br>• Identification of process bottlenecks<br>• Assessment of the need for institutionalizing processes<br>• Assessment of the need to consolidate processes and centralize data storage |

[a] Chad Guinea Worm Eradication Program.

Two interviews were conducted in French with the aid of an interpreter and the remaining interviews were conducted in English. Interviews were guided by semi-structured scripts that were tailored to a range of respondent roles (e.g., MoPH representatives, data managers, field staff, The Carter Center leadership). Questions probed about the simplicity of the surveillance system and the quality of the data [24]. For example, we requested explanations about how information is transferred at and between different levels of the system (i.e., village to field supervisors to regional hub to national level), with special attention to any potential bottlenecks. We asked if there were certain aspects of the surveillance system that repeatedly stimulate complaints from workers, require repeated training to maintain competency or required adjustments to problem solve. We also asked if information was captured at a specific-enough level of detail to support needed analyses, and if government officials, community leaders, and the general public trust the quality of the data for programmatic decisions. Findings from the interviews informed the development of mental models (described below) to reveal strengths and weaknesses of these processes.

## Mapping data elements and information flow

In order to prepare an inventory of the datasets and data management processes, we assessed the survey forms, which are the origination for all surveillance data. The 14 paper-based forms used to collect surveillance data are outlined in **Table 2**. The program may use ad hoc processes for collecting data related to the logistical processes. However, we focused on the

**Table 2. Description of survey forms used by CGWEP[a] to collect data.**

| Form number | Name of survey form | Objective | Brief description |
|---|---|---|---|
| 1 | Form for rumor of infection in animal | Surveillance | Documents rumors of infection in animals, one of the fundamentals of the surveillance structure |
| 2 | Form for investigating infections in animals | Surveillance | Documents cases in animals, including demographic and epidemiological data |
| 3 | Form for rumor of infection in humans | Surveillance | Documents rumors of infection in humans, one of the fundamentals of the surveillance structure |
| 4 | Form for investigating infections in humans | Surveillance | Documents case reports in humans with demographic and epidemiological data |
| 5 | Form for documenting rewards | Programmatic use | Tracks monetary rewards distributed to individuals reporting rumors that are cases |
| 6 | Form for burial practice of fish entrails in households | Programmatic use | Monitors sanitation practices that break the Guinea worm lifecycle |
| 7 | Form for burial practice of fish entrails in marketplaces | Programmatic use | Monitors sanitation practices that break the Guinea worm lifecycle |
| 8 | Monthly case tracking sheet | Summary report | Summarizes monthly cases; created by CGWEP[a] country office for all stakeholders |
| 9 | Summary form of data concerning Guinea worm control activities–health center level | Summary report | Summarizes monthly cases at a health center; created for CGWEP[a] for internal programmatic use |
| 10 | Abate (temephos) application form | Programmatic use | Guides calculation and treatment for larvicide application to a single waterbody to break the Guinea worm lifecycle and facilitates monitoring |
| 11 | Summary form of Abate (temephos) treatment | Programmatic use | Summarizes Abate (temephos) treatments completed each month for a specific coverage area |
| 12 | Form for evaluating village-level volunteers | Programmatic use | Evaluates standardized practice knowledge for village-level volunteers |
| 13 | Forms for supervisor visits | Programmatic use | Reports created by supervisors during field visits |
| 14 | Data summary form | Summary report | Summary reports; created by CGWEP[a] country office for all stakeholders |

[a] Chad Guinea Worm Eradication Program.

surveillance forms as these are routine processes and are the source of all the data related to Guinea worm epidemiology and eradication activities. We created data dictionaries for each survey form to document the type of data collected, the data management requirements, and the use of data by the program. Based on the consultations and the documents provided, we then mapped the data fields (i.e., variables) from each form to depict where and how information is gathered to understand how each component fit into the larger structure and goal of the organization.

## Mental models: Information value cycle and data-information system-context (DISC) rings

Mental models use visual representations to help identify, evaluate, and understand the specific elements and procedural knowledge for a specific process [25]. Based on the information we obtained from mapping the data fields, we used informatics evaluation mental models to identify inefficiencies, redundancies, and areas of improvement for data collection and management processes. Two frameworks were used to guide this process including the Information Value Cycle (IVC) and the Data-Information System-Context (DISC). We integrated the IVC and DISC (**Fig 1**) approaches to evaluate the information system because they address different system attributes.

The IVC, initially adapted from an early publication by Taylor et al, addresses the *function* of the information system [27] and the DISC deals with its *structure*; the combined approach provides a more holistic diagnosis of the entire information system in its current state and better elucidates system gaps.

The IVC is a mental model that helps visualize the functional value created iteratively from processes within an information system, revealing "information pathologies" of information processing that can be "diagnosed" and "treated" [26,27]. The IVC framework has been frequently used by other public health informatics use-cases for problem-solving activities [28,29]. We used the IVC to document the data collection, management requirements, and practice of the existing information system through the step by step perspective to portray a clear picture of some of the system's strengths and weaknesses [26]. The IVC contains six iterative steps, including *evaluate*, *plan*, *capture*, *manage*, *analyze*, and *use*. The *planning* and *evaluation* stages during the design and development phase of information systems can help identify opportunities for preventing problems that can occur later. The *planning* stage assesses organizational context, information needs, and information systems architecture. The *capture*,

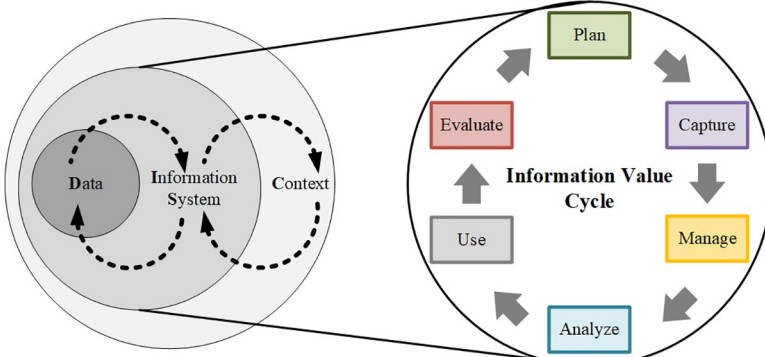

**Fig 1. Information Value Cycle—Data-Information System-Context (IVC-DISC) framework for contextual analysis matrix [26].**

*manage*, *analyze*, and *use* stages during the implementation and maintenance phase can identify opportunities for system improvements. The *capture* stage describes data collection, quality, type and format, and standards and the *manage* stage documents storage and retrieval, exchange, security, and integration practices. *Analyze* refers to assessment of data visualization, aggregation or linkage, and classification, and *Use* assesses how information system improves population health through situational awareness, aids in decision-making, and disseminates health information. Finally, the *evaluation* stage addresses the processes, outputs, outcomes, and impact of data collection.

The DISC [26] is another mental model used to characterize an information system's structure. The DISC framework recognizes the significance of the different personal, organizational, and environmental contexts that contribute to the success of an information system [30–32]. The goal of a public health program determines what *data* need to be collected. Data collection is facilitated and managed by the *information system* and the structure created to support this process. The *context* of this information system (e.g., health system, legislation, culture) determines its scope, scale, and design. A detailed assessment of the information system using the DISC framework can elucidate complex, cyclical relationships occurring between each ring. We mapped the concepts identified in the IVC to the DISC domains to create a contextual analysis matrix. This IVC-DISC framework has been frequently used to guide the solutions for various informatics problems in public health [28,29]. Concepts identified using the IVC were further categorized into granular contexts using the DISC framework. This IVC-DISC contextual analysis matrix reveals each concept identified in the IVC in the context of data, technology, people and process, organization, and environment. It is an evaluation model for assessing information systems, the value that they create, and their data flow. We used IVC-DISC framework as it breaks down contributors to the information system that are otherwise difficult to isolate. This allows for a holistic examination of these contributors and at the same time also allows for a measurable and quantifiable elicitation of these contributors and their roles in the system as a whole.

We reviewed data collection, reporting, and management practices to map data elements and document workflows, to provide insight into the data quality, and to identify areas for process improvement. IVC and DISC frameworks brought out the strengths and weaknesses of the information system and laid the foundation for streamlined processes.

## Results

### Data quality

Issues related to data quality were expressed by the interviewees. For example, community members (and potentially even field staff) may be inaccurately reporting data, especially canine containment data. The standard for containment requires people to tether their dogs at the blister stage, before worm emergence, but supervisors usually cannot arrive to the field location in time to verify that the dog was indeed tethered at the blister stage. Visible blisters must be promptly recognized so that dogs are quickly contained. Timing is critical for containment because there are often only a few hours between development of a blister and worm emergence, when the dog will contaminate water sources. A further challenge to surveillance is that Guinea worms in dogs are difficult to identify, especially before the worms have emerged, as blisters often occur near the paws or in between toes and are obscured by fur. Apart from these containment challenges with respect to timing and identification, paper-based data collection also jeopardized the quality of surveillance. Paper-based survey forms are prone to physical damage or loss and errors in handwritten information is a reality. Data could be inadvertently or maliciously changed, and auditing is more difficult because of the lack of version control.

### Data and information flow maps

Mapping data fields for each form revealed clear themes among the questions including location data, epidemiologic data about infections, and containment information, among others. **Fig 2** depicts the data fields mapped to their corresponding themes for the 'Form for investigating infections in animals' (form 2 in **Table 2**). Since the majority of Guinea worm cases in Chad occur in dogs, this form is used to collect the majority of the case-data in Chad [6]. Therefore, any system improvements involving this form significantly enhance the overall workflow.

Additionally, we mapped the data collection process and the flow of information for all forms to visualize the process of how and when data are collected, stored, managed, and utilized. As an example, **Fig 3** depicts the flow of data from rumor and infection investigation forms for humans (forms 3 and 4 in **Table 2**). Paper-based rumor investigation forms for humans (and animals) are completed in the field when staff learn of a rumor. The forms are then sent to the regional level for electronic data entry and cleaning, after which the data are shared and compiled at the headquarters-level. If a rumor is verified as a case by a CGWEP supervisor, data about the case and about reward dispersal (CGWEP awards cash incentives for community members to report suspected cases of Guinea worm in humans and animals) are collected via two additional forms. Thus, the mapping exercise revealed a total of three forms (using three unique Microsoft Excel databases) that could potentially derive from rumor investigations. Despite having a common origin, none of these databases are linked, and each is updated manually, thereby introducing the possibility of data entry error.

### Mental models: Information value cycle—data-information system-context (IVC-DISC) framework

The information collected, the people and processes, the tools used, and the outcomes of the CGWEP information system are represented in Fig 4. The figure also illustrates the current state and gaps in the information system. **Table 3** shows a contextual matrix combining the

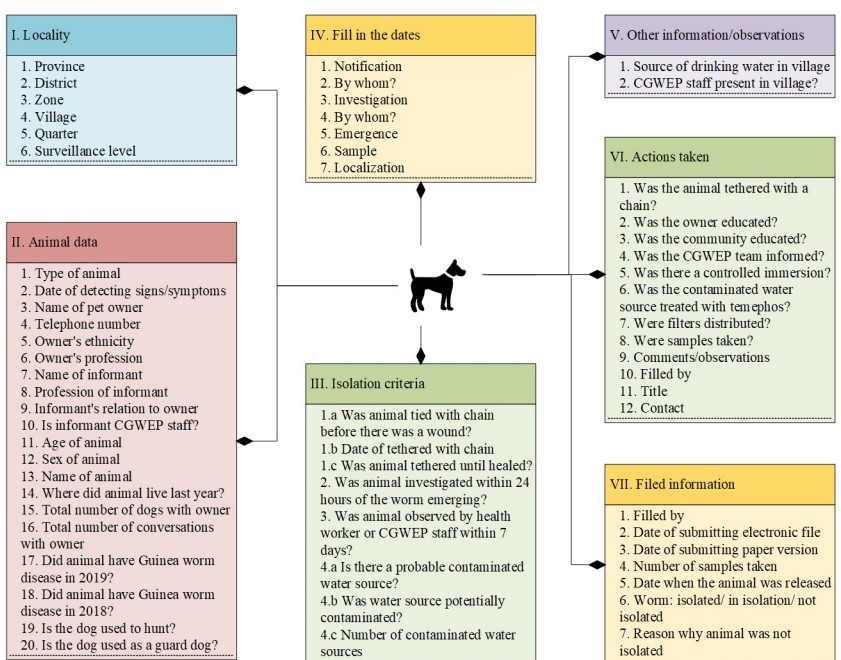

**Fig 2. Form for investigating infections in animals.**

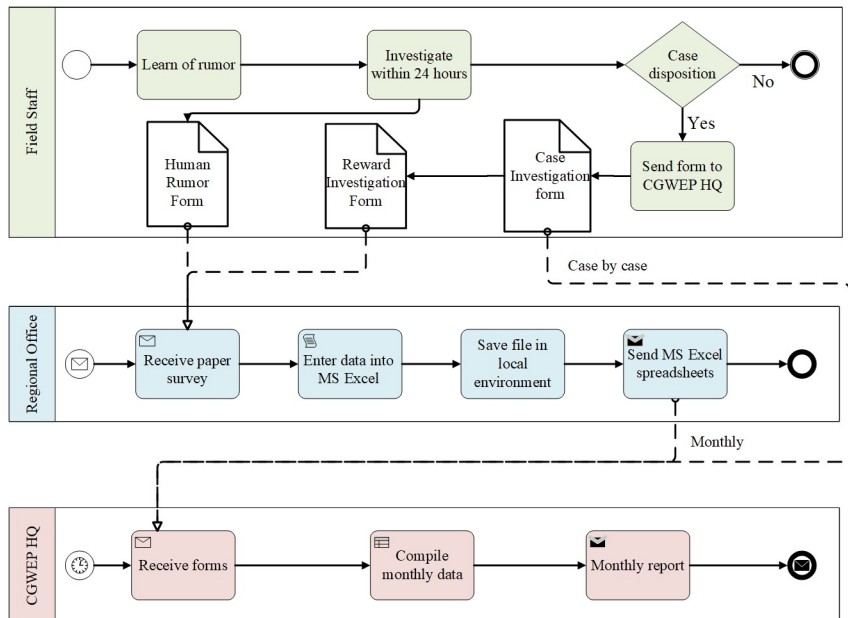

**Fig 3. Information workflow of rumor and infection investigations in humans.**

IVC and DISC models that describe each component in further detail. For example, concepts of data capture identified in the IVC were mapped to the DISC framework. This included the type of information collected, the technology used for data collection, the people and processes collecting the data, the organizational structure for data collection and use from the village-level to the international level, and the environmental factors affecting these processes.

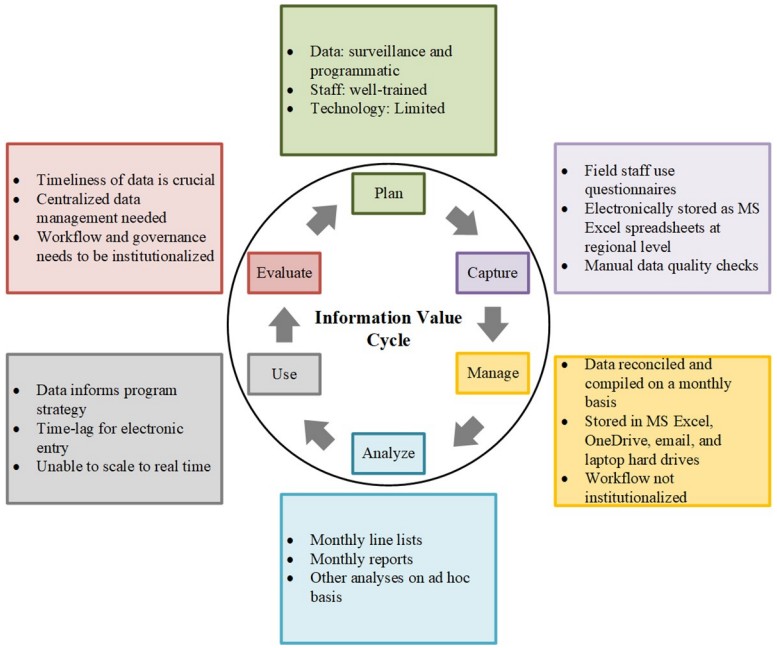

**Fig 4. Information Value Cycle representing the information collected, people and processes involved, tools, and outcomes of the information system.**

**Table 3. Dimensions of an information system: the IVC-DISC contextual analysis matrix.**

| | | Dimensions of an Information System | | | | |
|---|---|---|---|---|---|---|
| | | Components of the DISC framework | | | | |
| | | Data | Information System | | Context | |
| | | Information | Technology | People & Process | Organization | Environment |
| Stages of the Information Value Cycle | Plan | Data regarding 'rumors', cases, case containment, chemical treatment of water bodies, other programmatic data | Limited network connection, use of personal and official laptops, reliance on paper-based data capture | Well-trained staff at all levels, workflows and processes need to be developed and implemented | All stakeholder organizations–, MoPH[a], CDC[b], WHO[c], TCC[d] - are motivated to solve the problem | Field locations can be remote, resources reflect the level of surveillance and can be difficult to source. |
| | Capture | Field staff collect data via paper forms and surveys, regional staff enter data electronically | Field staff use paper-based surveys and forms, data are entered into Microsoft Excel | Field staff, technical advisors, data entry staff, and data manager | Village level paper-based surveys and forms, electronically entered at the regional and national level | Some turnaround time between manual data collection and electronic data capture |
| | Manage | Data reconciled and compiled at the national level, data quality is also checked manually | Microsoft Excel, email, laptops, and OneDrive are used for capturing and storing data | Data quality manually checked at all levels, need to institutionalize workflows | Data managed at the discretion and technical skill level of the data manager | No centralized data collection platform, data not linked to each other |
| | Analyze | Line lists compiled and reports created monthly, additional analyses conducted on ad hoc basis | Periodic reporting done; ad hoc analyses conducted; no centralized dashboard | Well-trained staff to conduct analyses, monthly reports could be generated with more automation | TCC[d] shares monthly reports to CDC[b], MoPH[a], WHO[c], has appetite for complex ad hoc analyses | Analysis must be conducted ad hoc, as there is no metadata, data dictionary, or a data governance policy |
| | Use | Information is crucial to inform surveillance and intervention strategies; good data is critical for program success | There is a time-lag for data to be available electronically; prevents a scalable, program level intervention based solely on the data | Active and passive surveillance data, including 'rumors', are crucial and heavily relied on by field staff to prevent disease transmission | CGWEP[e] relies heavily on the data for prevention of disease transmission and to form strategies to combat the disease | Simultaneous surveillance of humans and animals, including monitoring of the ecosystem; Data of great importance |
| | Evaluate | TCC[d] is a highly scientific agency and uses evidence to drive policy | Centralized data collection and storage seems to be a strong demand amongst the field staff | Staff favor electronic data for higher quality data; centralized data repository will streamline many processes | TCC[d] is modernizing IT infrastructure; CGWEP[e] needs a robust infrastructure to capture accurate data and make it available in a timely manner | Organization level and field level feedback seem to favor some form of centralized platform for data collection and storage |

[a] Ministry of Public Health and National Solidarity.

[b] Centers for Disease Control and Prevention.

[c] World Health Organization.

[d] The Carter Center.

[e] Chad Guinea Worm Eradication Program.

Similarly, this was done for all other functions of the information system. Contextualizing components of information systems quantified measurable potential changes that can be introduced to streamline the process and improve efficiency.

The results of the IVC by each of the six themes are described below.

*Plan*: CGWEP collected both programmatic data and epidemiologic surveillance data. The staff reported having received adequate training in surveillance and program activities, but they faced challenges in technology adoption due to limited connectivity in the region. *Capture*: Data were first collected in paper-based surveillance forms by field staff. These were electronically entered via Microsoft Excel and were stored in separate locations including CGWEP issued laptops, The Carter Center's cloud-based storage platform, personal laptops, and email folders. Although quality checks were regularly performed on data collected for an individual form, manual quality checks were only occasionally performed across spreadsheets. *Manage*:

Data were reconciled and compiled into reports on a monthly basis. Data were stored in different locations and transported using multiple methods such as email and flash drive transfer within Chad when there was no or limited internet access. There were no standardized workflows for these processes. *Analyze*: Monthly reports were compiled and reported to partner agencies. Ad hoc analyses and reports were created in separate tabs on Microsoft Excel spreadsheets. More detailed and longitudinal analyses for reports and other publications were regularly conducted by program epidemiologists using statistical software. *Use*: Data were fundamental for informing strategies for case containment, health promotion, and provisioning safe drinking water. There was some time lag for producing actionable data due to paper-based data collection and management practice. *Evaluate*: Accurate, timely data are required for continued program success. Several processes were identified that could improve data collection and management, including more centralized storage practices and streamlining of workflows systems.

## Discussion

Results from this project showed that there were several data and system inefficiencies, as well as practical opportunities to resolve them (**Table 4**). For example, the CGWEP data were collected on numerous (14) paper-based forms, which were electronically converted to spreadsheets, and ultimately used to periodically compile reports. Reconciling surveillance data and compiling periodic reports consumed significant time and resources. Further, the processes of collecting and storing information were not institutionalized and required significant manual labor and valuable time. The information was stored in different locations, often in vulnerable physical storage devices. To address these issues, we proposed a four-phased, incremental roadmap to implement an integrated public health information and surveillance system that includes agreements regarding data management, such as accessing and sharing data, data standards, procedures to reduce data collection burdens, and communication practices for the system [33]. We distill each key issue faced by the information system in **Table 4** and further describe how these are addressed by a multi-phase approach of system improvements. An incremental approach that increases in complexity over time will create a strong foundation of data management practices and ensure successful implementation and adoption. The proposed roadmap also provides opportunities for testing and troubleshooting each phase of the implementation, helping to assure that there is progressive improvement of the surveillance system that is both sustainable and a "good fit" for the program. This proposal is also an example of an informatics quality improvement roadmap that could serve as a model for other NTD prevention and control programs.

### Phase I: Develop a data governance plan

Although standard operating procedures were in place describing field work (e.g., collecting worm specimens, treating water sources with temephos), no detailed guidance existed to explicitly describe data collection and management processes. We proposed development of a data governance plan that includes clearly defined workflows for data collection and management, with delineated roles and responsibilities for each stakeholder (**Table 4**). The data governance plan should also address issues such as standardized data storage practices and clear naming conventions for data files and folders to improve accessibility and compliance with information security best practices. Additionally, the plan should include maintaining a centralized, up-to-date data dictionary for all users and for all forms to improve and promote consistency in use and reuse of data, thus saving time during data analyses and interpretation. A data governance plan also promotes institutional memory and shortens the learning curve for incoming staff. Importantly, this plan does not require intensive resource allocation for

**Table 4. Key issues facing the Chad Guinea Worm Eradication Program information system, how these were addressed, and potential challenges.**

| Key issues discovered | Corresponding phase | Implementation requirements | Barriers |
|---|---|---|---|
| Data collected through paper-based surveys | • Phase III: Electronic data collection platform | • Electronic devices (e.g. mobile phone)<br>• Data collection tool (software) | • Resources[a] for purchasing phones, configuring, and maintaining the information system<br>• Reliable telephone/internet network |
| Manual data collection and data entry (digitization) process | • Phase III: Electronic data collection platform | • Electronic devices (e.g. mobile phone)<br>• Data collection tool (software) | • Resources[a] for purchasing phones, configuring, and maintaining the information system<br>• Reliable telephone/internet network |
| | • Phase II: Electronic data entry platform, collated to central database | • Electronic data entry, collated to central database | |
| Data stored in physical devices (laptops, hard drives, etc.) | • Phase II: Centralized data storage | • Centralized database | • Resources[a] for purchasing and maintaining the database and information system |
| Data reconciled and compiled periodically | • Phase I: Data governance strategy, centralized data dictionary, and institutionalized workflows | • Dedicated time for data governance activities | • Data governance strategies must be practiced and institutionalized |
| | • Phase II: Electronic data entry platform, collated to central database | • Standardized data elements and format | None |
| | • Phase III: Linking data at its source to eliminate need for reconciling data | • The data collection tool to link data during collection | • Resources[a] for purchasing and maintaining cloud-based system<br>• Reliable telephone/internet network |
| Data management dependent on the technical skills of the team | • Phase I: Data governance strategy, institutionalized workflow | • Well-designed and 'good fit' workflows | • Data governance strategies must be practiced and institutionalized |
| | • Phase II: Centralized, relational database | • Standardized data management techniques | • Resources[a] for purchasing and maintaining the database |
| No metadata, data dictionary, or a data governance strategy | • Phase I: Data governance strategy, metadata, and central data dictionary | • Manually created to serve as the single source of truth | • Needs regular and periodic update to reflect accurate information |
| | • Phase II: Database-generated data dictionary and metadata | • Centralized database | Resources[a] for purchasing and maintaining the database |
| Data stored in disparate locations; data not linked to each other. Time lag for data to be available. | • Phase II: Electronic data entry platform, collated to central database | • A centralized database, data stored in one place | • Resources[a] for purchasing and maintaining the database |
| | • Phase III: Linking data at its source to minimize redundancies and duplications | • The data collection tool to link data during collection<br>• The platform will remove duplicate, overlapping data fields from data collection | • Resources[a] for configuring bidirectional information exchange between data collection device and central database<br>• Reliable telephone/internet network |
| | • Phase IV: Data available near real-time | • Cloud-based, integrated electronic devices and database | • Resources[a] for purchasing and maintaining cloud-based system<br>• Reliable telephone/internet network |
| Manually created recurrent reports and analyses. | • Phases I & II: Standardized data, updated metadata, central database | • Templates for recurrent reports<br>• Metadata for ad hoc analyses | Resources[a] for purchasing and maintaining the database |
| | • Phase IV: Data available near real-time | • Cloud-enabled devices<br>• Dashboards to replace recurrent reports | • Resources[a] for configuring cloud-based bidirectional information exchange between devices |

[a]Resource utilization can only be estimated; actual cost and timeline will vary on local factors and context.

Key issues identified are listed in the order in which they appear in the workflow from data collection, to data management, to information dissemination

development or implementation, and ultimately saves both time and resources [34]. Therefore, this Phase in the roadmap could easily be adapted to any disease control, elimination, or eradication program that relies on surveillance data.

## Phase II: Transition to electronic data entry and centralized data storage

For Phase II, we proposed adopting an electronic data entry platform (**Table 4**). For CGWEP, we proposed NEMO/ELMO, an open-source and cloud-based mobile data collection and

reporting system developed by The Carter Center, because the platform is familiar to field staff [35,36]. This would permit paper-based data collection at the field-level while allowing data to be stored centrally in NEMO/ELMO's SQL database, thereby eliminating the need to enter data onto spreadsheets and mitigating risks arising from data residing in different physical devices and reduce loss of data due to stolen, lost, or damaged equipment. Electronic data entry and centralized storage would automate reconciliation and compilation of data (e.g., data cleaning, integration, and performing quality checks), which are currently being done manually. For example, data quality measures such as standardized lists (drop-down lists) could be developed for categorical variables (e.g., sex, village, etc.); these would ensure that data are entered in a standard format without spelling mistakes. To test this process, as part of this Info-Aid, we built survey instruments in NEMO/ELMO for the 'form for rumor of infection in animals' and 'form for investigating infections in animals' (forms 1 and 2 in **Table 2**).

NEMO/ELMO is hosted as a cloud platform and therefore can be accessed in near real-time by staff who have appropriate permissions. The data managers and analysts would not need to maintain data dictionaries because they would be automatically generated by NEMO/ELMO.

Resources required to implement this phase include (but are not limited to) purchase of centralized storage system/service and staff dedicated to configuring and maintaining this database. Some NTD programs already use electronic data entry and centralized storage systems. For example, the Mission Rabies App has been used to collect information about dog vaccinations, rabies community education, and field surveys in 16 countries [37].

## Phase III: Transition to complete electronic data collection and management via a relational database

For Phase III, we proposed that NEMO/ELMO would be used for all data *collection* in the field (rather than only electronic data *entry*), with data collated into an inbuilt relational database (**Table 4**). All survey forms would be made available on mobile devices and tablets, completely replacing paper-based surveys. NEMO/ELMO has capability beyond just collecting data via mobile devices and could potentially also be used to populate data and additional information onto the mobile device itself. Our results showed that the core surveillance data forms (forms 1–4 in **Table 2**) collect some similar and repetitive data elements in multiple survey forms. For example, **Fig 2** illustrated all the data fields collected in the 'form for investigating infections in animals' (form 2 in **Table 2**) such as demographic information (e.g., name, address, ethnicity, profession), information about the animal (e.g., animal type, age, sex, previous infection), case containment information (e.g., animal tethered with lock and chain, contaminated water source). We organized the data fields into thematic sections reflected in the form. Similar mapping exercises with other forms revealed a great deal of overlap in the type of data collected with other forms within corresponding sections, particularly in the demographic data. The system in its current state provides no opportunity to overcome this redundancy due to use of paper-based surveys. Making data available in the centralized database to the NEMO/ELMO enabled mobile devices used for data collection in the field will allow CGWEP to link the data at its source at the time of data collection. Additional resources for electronic data collection would include the purchase of electronic devices (such as mobile phones) and staff dedicated to programming the devices.

Data available in NEMO/ELMO's inbuilt relational database, if made available to the device collecting data in the field, would allow linking of data points to the same individual, animal, household, or village using a system-generated unique identifier. After entering the case-patient (or dog owner) name and demographic information, NEMO/ELMO could be

programmed to identify a match, if it existed in the database. The field staff administering the survey would then have the ultimate say over whether two records were indeed a match. If a match, then the remaining variables could be automatically populated in the survey form. The field staff could verify the contents for accuracy, with manual override for any discrepancies. This linking of data, and the ability to validate the data in real-time in the field would greatly increase the quality of the data by reducing redundancy and minimizing error associated with repeated data entry. We recommended that Phases I and II be used to prepare the foundation for an improved data capture and management process. Implementing electronic data capture at the source of the information is likely to significantly improve the timeliness and accuracy while reducing redundancies in data management practice. This bidirectional information exchange, however, would require significant technology and human resources to manage the system, which may not be available to many NTD programs.

### Phase IV: Cloud-based integration with enterprise IT infrastructure

In Phase IV, we proposed to integrate the steps in Phases I–III together with existing enterprise IT infrastructure. Namely, we proposed to house CGWEP data in a centralized cloud repository and promote exchange of data with unambiguous, shared meaning. This would make the data easily accessible by multiple parties (e.g., MoPH, CDC, WHO) in near real-time with centralized access controls and privileges management (**Table 4**). If carried out, recurring analyses could be displayed in a data dashboard and could reduce the burden of monthly reporting. Dynamic analysis and visualization of data has potential to reveal near real-time epidemiological patterns as they evolve. Most importantly, all these data and insights could be available to the field staff who are responding to a rumor or are investigating a case and thereby improving public health response capacity. The NTD Mapping Tool, for example, also provides dynamic analyses and visualizations by integrating NTD data onto a cloud platform and reduces the burden for reporting and provides a tool for planning, implementation, and evaluation of NTD control activities [38].

### Limitations

Field staff that are responsible for data collection were not interviewed for this project, as we instead focused our information gathering on GCWEP and Carter Center senior-level office staff based in N'Djamena and Atlanta (Table 1). This could bias our findings by omitting important weaknesses in the information system only visible to GCWEP members that collect data on the ground. The phased plan we propose here may not be suitable for all NTD programs because of the financial and human resources required and because of connectivity issues in remote areas that may hinder the ability to transmit and receive data. In the process of this transition CGWEP did incur costs related to monthly data transfer and storage, the cost of purchasing 700 cell phones to facilitate the use of the data collection platform, and person-hours devoted to configuring devices and managing incoming data.

### Conclusion

This investigation resulted in identification of system level problems and potential solutions. Although informatics assessments have been conducted on macrolevels for infectious disease [39] and surveillance evaluations performed for Guinea worm disease [4,20,40], our approach is novel in that we assessed the information system from the surveillance need, data management, and data use perspectives for Guinea worm disease. The framework we used here could be adapted by other programs looking to assess their information systems to support surveillance and programmatic needs.

## Acknowledgments

We thank the CGWEP staff for participating in this project. Giovanna Steel and Karmen Unterwegner from The Carter Center also provided helpful insights throughout the duration of this project. The findings and conclusions in this report are those of the authors and do not necessarily represent the official position of the U.S. Department of Health and Human Services, the Centers for Disease Control and Prevention, or the authors' affiliated institutions.

## Author Contributions

**Conceptualization:** Saugat Karki, Adam Weiss, Jina Dcruz, Sarah Yerian, Sharon L. Roy, Mary L. Kamb, Sarah Anne J. Guagliardo.

**Data curation:** Saugat Karki, Philip Tchindebet Ouakou, Elisabeth Chop, Hubert Zirimwabagabo, Sarah Anne J. Guagliardo.

**Formal analysis:** Saugat Karki, Jina Dcruz, Beth L. Rubenstein, Sarah Anne J. Guagliardo.

**Investigation:** Saugat Karki, Jina Dcruz, Dorothy Hunt, Brandon Haigood, Elisabeth Chop, Beth L. Rubenstein, Sarah Anne J. Guagliardo.

**Methodology:** Saugat Karki, Adam Weiss, Jina Dcruz, Sarah Yerian, Sharon L. Roy, Mary L. Kamb, Sarah Anne J. Guagliardo.

**Project administration:** Saugat Karki, Adam Weiss, Jina Dcruz, Hubert Zirimwabagabo, Sarah Yerian, Sharon L. Roy, Mary L. Kamb, Sarah Anne J. Guagliardo.

**Resources:** Adam Weiss, Jina Dcruz, Dorothy Hunt, Philip Tchindebet Ouakou, Elisabeth Chop, Hubert Zirimwabagabo, Sarah Anne J. Guagliardo.

**Software:** Saugat Karki, Dorothy Hunt, Brandon Haigood.

**Supervision:** Adam Weiss, Jina Dcruz, Sarah Yerian, Sharon L. Roy, Mary L. Kamb, Sarah Anne J. Guagliardo.

**Validation:** Saugat Karki, Adam Weiss, Dorothy Hunt, Philip Tchindebet Ouakou, Elisabeth Chop, Hubert Zirimwabagabo, Sarah Yerian, Sharon L. Roy, Mary L. Kamb.

**Visualization:** Saugat Karki, Sarah Anne J. Guagliardo.

**Writing – original draft:** Saugat Karki, Adam Weiss, Jina Dcruz, Beth L. Rubenstein, Sharon L. Roy, Mary L. Kamb, Sarah Anne J. Guagliardo.

**Writing – review & editing:** Saugat Karki, Adam Weiss, Jina Dcruz, Dorothy Hunt, Brandon Haigood, Philip Tchindebet Ouakou, Elisabeth Chop, Hubert Zirimwabagabo, Beth L. Rubenstein, Sarah Yerian, Sharon L. Roy, Mary L. Kamb, Sarah Anne J. Guagliardo.

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
