## [Decision Letter · Decision Letter 0]

13 Apr 2021

Dear Dr. Karki,

Thank you very much for submitting your manuscript "Assessment of the Chad Guinea Worm Surveillance Information System: A Pivotal Foundation for Eradication" for consideration at PLOS Neglected Tropical Diseases. As with all papers reviewed by the journal, your manuscript was reviewed by members of the editorial board and by several independent reviewers. In light of the reviews (below this email), we would like to invite the resubmission of a significantly-revised version that takes into account the reviewers' comments. 

We cannot make any decision about publication until we have seen the revised manuscript and your response to the reviewers' comments. Your revised manuscript is also likely to be sent to reviewers for further evaluation.

Sincerely,

Zvi Bentwich, M.D

Associate Editor

Achim Hoerauf

Deputy Editor

Reviewer's Responses to Questions

**Key Review Criteria Required for Acceptance?**

**Methods**

-Are the objectives of the study clearly articulated with a clear testable hypothesis stated?

-Is the study design appropriate to address the stated objectives?

-Is the population clearly described and appropriate for the hypothesis being tested?

-Is the sample size sufficient to ensure adequate power to address the hypothesis being tested?

-Were correct statistical analysis used to support conclusions?

-Are there concerns about ethical or regulatory requirements being met?

Reviewer #1: The methods are structured less than effectively. Two of the earlier sections (Initial description, Mental models) might perhaps better placed in the Introduction, as they describe the starting place. That said, this section also would be hard to understand for someone who was not already familiar with the GW eradication program or very similar efforts. A further step back is required to understand the structures and practices of the program, and particularly how these use, or do not use, data collection and analysis.

While the Methods provide an outline of the approach, they are less clear on the analytical process. 

Respondents: While the respondents were asked for consent, was informed consent given, and was it verbal or written? Which authority granted ethical approval for the study? What form did the consultation take? Face-to-face interviews? How many respondents of each type were there? This is important to understand how the practices were sampled. Some of this information is provided in a Supplementary Table, but this does not appear to be cited in the paper text, and so comes as a surprise at the end. This Table could be included in the main paper. 

This Table suggests that there is a heavy bias in sampling towards fairly senior, and office-based, staff. The most "junior" are data entry staff and Technical Advisors, also all based in the Chad country office. An important point, therefore, is that no true "field staff" i.e. those who actually collect the data appear to have been sampled. Which seems like a serious omission in terms of understanding data generation and subsequent flows.

Reviewer #2: This paper is very weak on identifying related work, research questions and research methods. see below

Reviewer #3: Partially. This paper is a report of a mental map to evaluate the data collection processes of the CGWEP. There are no real data included and no statistical analyses.

**Results**

-Does the analysis presented match the analysis plan?

-Are the results clearly and completely presented?

-Are the figures (Tables, Images) of sufficient quality for clarity?

Reviewer #1: Table 1 feels more like Results than Methods. Are forms 10 and 11 the same, as their description differs in only 1 word "Systematic"? Perhaps their function/content could be distinguished better? 

I am actually surprised there are as few as 15 forms, which suggests a good deal of additional ad hoc data collection and distribution that covers processes such as collection of worms for despatch for identification, counts of animals during censuses, documentation of village visits by a range of voluntary and paid staff, and other programmatic activities. 

Because of the lack of detail in the Methods, the first part of the results are a surprise as they deal with very specific challenges to data collection and recording (not recording dog containment), rather than the information systems. This elicitation of problems is not addressed in any detail in the Methods, which deal primarily with structural aspects. This specific aspect (dog containment) is then discussed in some detail, which again seems out of place here. Were no other aspects of the program considered equally challenging? 

There follow no further results relating to data content or quality, which again is surprising. It does not feel as though the data requirements, for actual control, are at all well explored with the respondents. Instead, the analysis is largely based on the forms and their overlapping content and functions. Are the authors therefore content that all the necessary information is being collected and that the only improvement is in how these data are handled?

Similarly, surveillance sensitivity (L163) is not described specifically. And is not, in itself directly related to informatics processes, although it is certainly of importance. 

L188-199 are again Methodological in tone and content, while the lines 193-197 are not easily understood and their purpose is not clear. it is not clear how this analytical outcome arises from the practical work undertaken. This highlights the missing content that would allow these components to be joined in the mind of the reader.

L213-214. This is the first mention of the use of program data in epidemiological or other research activities. This might be expanded and made more prominent throughout the paper, as this is an additional vital use of program information.

Reviewer #2: see below

Reviewer #3: Yes

**Conclusions**

-Are the conclusions supported by the data presented?

-Are the limitations of analysis clearly described?

-Do the authors discuss how these data can be helpful to advance our understanding of the topic under study?

-Is public health relevance addressed?

Reviewer #1: While the conclusions are valuable and supported by the study, there is little express consideration of the constraints acting upon the roadmap, which are likely of considerable general relevance to NTD control programs. How might this roadmap be implemented in a timely way in the circumstances found in Chad, and other similar settings?

The Conclusions jump straight into the recommended road map, but do not initially distil the key issues faced by the program, as revealed by the preceding analyses.

Reviewer #2: see below

Reviewer #3: Yes

**Editorial and Data Presentation Modifications?**

Reviewer #1: Some minor editorial points and some points of biological specificity that might be refined...

L38 "involves a freshwater copepod (tiny crustacean) in its lifecycle" is an oddly informal detail for the first line of the article - perhaps this detail might be framed more biologically and moved elsewhere in the introduction.

L43 It has been a convention that the term "cases" refers only to humans, while animals are termed "infections". Some people (not me particularly) adhere with some passion to this distinction. Given the prominence of this change in vocabulary, in the early stages of the Introduction, the authors might want to reflect on whether this is a deliberate and considered change.

L43 Perhaps the 2020 numbers might be used now that they are available.

L51 Given its use to control crustaceans in this context, temephos is perhaps not best described as an insecticide. It is often described as an organophosphate larvicide or similar.

L52 More strictly, the filters yield water that is free of copepods that might carry Guinea worm larvae

L57 I would not consider relational databases, or Excel / Access to be recent technological advances. Clearly there are some highly relevant advances that are relatively recent, such as Cloud-based systems. Just tweak the text a bit here. Some tools are only basic and it will come as a surprise that there is such a reliance on paper record keeping, given how long established basic tools are. Moreover, the paper advocates moving away from simple but inefficient systems like Excel spreadsheets.

L61 Slight hyperbole about a tipping point in history, perhaps?

L81 This line is contradictory, in that it identifies historical data collection practices *at the time* of the study. Which is it? Is this what the investigators found, or is it how they understood things were in the past? Perhaps this needs to be stated explicitly, by giving time frames?

L83 individual "host"? Is this asking for linkages between disparate information about the same host/worm to be linked, or for information about the host and the worm to be linked, or both?

L149 The Results start with some unnecessary recapitulation of the Methods.

L262 I think this is the first mention of NEMO/ELMO, which requires some explanation, as it then dominates the remaining recommendations.

Reviewer #2: see below

Reviewer #3: None

**Summary and General Comments**

Reviewer #1: This is potentially a valuable account of the data and information operations of a program with global significance. It is of general interest to other disease control and eradication programs and, in principle, is within scope for the journal. 

The account of information flow and challenges to effective use of data is pointed, clear and useful.

That said, in parts this manuscript reads rather more like a contract report than a research paper and the tone and structure needs some attention to bring it up to the latter standard. Broader context from the literature in this area is largely lacking. 

There are some key issues that I think should be addressed before publication, that would greatly improve the value of the paper. 

The Introduction is brief and highly specific, giving some basic details of the GWEP in Chad and of the basics of informatics analyses. However, there is scope to introduce more broadly the important role of informatics, and improvement therein, to disease eradication and control, including successful programs and those that are currently struggling. Some contrast between the specifics of public health informatics in challenging environments found in low to middle income countries, such as are found in the remaining GW endemic countries, compared to high income countries would be valuable. 

While it is clear that the IVC-DISC approach allows some formal analysis of the informatics challenges, these are not set in context or analysed in any obvious way, that would then substantiate the roadmap towards improvement. While some of the informatics shortcomings/failings are clear (too many paper forms, clunky data handling, data not well connected, risk of data loss), it is not substantiated or made clear how great the benefit would be if this were fixed. It is not specifically addressed how such flawed systems have nevertheless worked so well up to this point, and/or how indispensable improvements are at this stage. Effectively, is this essential works at this point, or is it "nice to have"? 

Proposals, e.g. for a data governance plan, are introduced with little substantiation of options beyond "contractor-type" recommendations. These recommendations then stop short of explicit content, requirements and implications. These could perhaps be tabulated to be efficient. This section is particularly suggestive of a contractor report, as opposed to a research paper. However, it again stops short of spelling out the implications in terms of cost, scale, timelines for the recommended roadmap, and the context of a plan for disease elimination in Chad that currently measures out over only a few years.

Finally, and perhaps most importantly, the paper misses an absolutely key point in relation to the specificities of *eradication* and of a disease with human and animal hosts, as opposed to effective control. To identify the importance of informatics to eradication, and certification, is a key requirement of the Guinea worm program, that generalises to other eradication programs, and yet this specificity is not mentioned at all anywhere in the paper. Neither does the paper highlight the specific challenge facing Guinea worm eradication of the involvement of people and animals, and any informatics challenges this generates.

Reviewer #2: This paper talks about an effort of assessing the Chad Guinea Worm Surveillance Information System. The authors used Information Value Cycle and Data Information System-Context framework to understand the information generated and identify the gaps. 

As the paper stated, this topic is very important and the outcome of the information system has great social impacts, so this is very important work. However, since this manuscript is marked as a research paper, my review will focus on the innovation of the research questions and the validity of the research method. 

Firstly, there is no literature of the related work. There probably only one Guinea Worm Surveillance Information System in Chad, but there must be related work on examining the validity and the gap of an information system for diseases. Consequently, there should be a related work section to indicate the limitations of our scientific understanding of the problems, and the innovation of this paper. 

Secondly, there are no research questions presented in the paper. Consequently, it is not clear the focus of this paper, and the set of specific issues or questions that the authors want to address in this paper. As a result, it is not clear whether or not the results that the authors presented in this paper are the important ones and/or the comprehensive list of ones. 

Thirdly, there is a brief introduction of the two mental models used in this paper, and a reference is given for the two models, which is a book chapter. However, there is no further justification on the literature around these two models. Are they well known models in the field? Have other related work used these two models and obtained insight results? Yes, these two models helps to identify the functions and structure of the system, but why are these two aspects important for understanding the information generated and the gap that the authors seek? Particularly, the gap can exist at any part of the information system, so how do these two models help to identify the gap?

The four phases presented in the discussion section sound very important, however, it is not clear to me how these phases are mapped to the results presented in the result section.

Overall, this is a very important project report, but it is not a high quality research paper to be published on a PLOS journal yet.

Reviewer #3: This is a rather unconventional submission for PLOS NTDs, as it does not contain any research, data or data analysis. The manuscript reports an analysis of the current data recording procedures used by the Chad Guinea Worm eradication program (CGWEP), and provides recommendations for how the program might improve the speed, accuracy and efficiency of their data collection and analysis activities. The CGWEP collects data of provisional and confirmed cases using four different forms, each of which contains a large number of questions that need to be completed. The authors conducted an analysis of the forms and the data collection and analysis process using a combination of two mental model frameworks and identified a number of redundancies and potential weak points in the current process. Based upon this, they proposed a four-step process to allow the CGWEP to move to a cloud-based data capture and analysis platform.

1. The authors propose a gradual process of migrating the current paper-based system to a cloud based central database. While it is obvious that this will increase the timeliness of data reporting, reduce data entry and transcription errors and will result in a central repository for all data, the authors do not provide any estimate of the cost of the cloud conversion. How many data entry devices (cell phones tablets, etc) will be needed to effect this conversion? Will this be affordable? How daunting are the connectivity issues for data upload in Chad likely to be?

2. The authors suggest that this process might be a useful guide for other disease elimination programs to follow to move their data systems to the cloud as well. I agree that this may be very useful. But most of the NTD elimination programs are not nearly as advanced as the Guinea Worm program, and as such, they employ many more people. For example, the river blindness elimination programs in Africa all rely upon volunteer community drug distributors (CDDs) to distribute the needed drugs and to keep the records of who has been treated and who has not. Each community has a number of CDDs – usually at least one per kinship group. Currently, all the treatment records are kept by th CDs on paper. It would seem to me to be very expensive to try to migrate the CDD collected data to a cloud-based platform. I would like to see the authors expand further on the generality of what they have done here to other PC NTDs.

3. One suggestion that the authors had concerning the advantage of an electronic data entry platform was that the platform could be programmed to auto fill much of the data on a given individual, saving the person entering the data time. But in the case of the CGWEP, the number of cases that are being reported is quite small, as the infection is near elimination. Given the relative rarity of the infection, how often does the program field case reports from the same individual more than once?

PLOS authors have the option to publish the peer review history of their article (what does this mean?). If published, this will include your full peer review and any attached files.

Reviewer #1: No

Reviewer #2: No

Reviewer #3: No
---

## [Editor Report · Decision Letter 1]

23 Jul 2021

Dear Dr. Karki,

We are pleased to inform you that your manuscript 'Assessment of the Chad Guinea Worm Surveillance Information System: A Pivotal Foundation for Eradication' has been provisionally accepted for publication in PLOS Neglected Tropical Diseases.

Best regards,

Zvi Bentwich, M.D

Associate Editor

Achim Hoerauf

Deputy Editor

---

## [Editor Report · Acceptance letter]

4 Aug 2021

Dear Dr. Karki,

We are delighted to inform you that your manuscript, "Assessment of the Chad Guinea Worm Surveillance Information System: A Pivotal Foundation for Eradication," has been formally accepted for publication in PLOS Neglected Tropical Diseases.

Best regards,

Shaden Kamhawi

co-Editor-in-Chief

Paul Brindley

co-Editor-in-Chief
